Altered O-linked glycosylation in benign and malignant meningiomas

Talabnin Chutima 1
Trasaktaweesakul Thanawat 2
Jaturutthaweechot Pitchanun 1
Asavaritikrai Pundit 3
Kongnawakun Dusit 4
Silsirivanit Atit 5
Araki Norie 6
Talabnin Krajang 4 krajang.t@sut.ac.th
1 School of Chemistry, Institute of Science, Suranaree University of Technology , Nakhon Ratchasima , Thailand
2 School of Translational Medicine, Institute of Medicine, Suranaree University of Technology , Nakhon Ratchasima , Thailand
3 School of Surgery, Institute of Medicine, Suranaree University of Technology , Nakhon Ratchasima , Thailand
4 School of Pathology, Institute of Medicine, Suranaree University of Technology , Nakhon Ratchasima , Thailand
5 Department of Biochemistry, Faculty of Medicine, Khon Kaen University , Khon Kaen , Thailand
6 Department of Tumor Genetics and Biology, Graduate School of Medical Sciences, Faculty of Life Sciences, Kumamoto University , Kumamoto , Japan
Sistla Srinivas
Electronic publication date: 2024 Jan 22
Publication date: 2024
Volume: 12
Electronic Location ID: e16785
Received 2023 Aug 20; Accepted 2023 Dec 19
Copyright: © 2024 Talabnin et al.
Copyright year: 2024
Copyright holder: Talabnin et al.
License: This is an open access article distributed under the terms of the Creative Commons Attribution License, which permits unrestricted use, distribution, reproduction and adaptation in any medium and for any purpose provided that it is properly attributed. For attribution, the original author(s), title, publication source (PeerJ) and either DOI or URL of the article must be cited.
License URL: https://creativecommons.org/licenses/by/4.0/

Keywords: Meningiomas, O-linked glycosylation, Sialyltransferases, Fucosyltransferases, Glycosyltransferases, Mucin

Funding: Suranaree University of Technology Thailand Science Research and Innovation (TSRI) National Science Research and Innovation Fund (NSRF) 160335 The work was supported by the Suranaree University of Technology, Thailand Science Research and Innovation (TSRI) and the National Science Research and Innovation Fund (NSRF) (NRIIS number 160335). The funders had no role in study design, data collection and analysis, decision to publish, or preparation of the manuscript.

==============================
Background

Changes in protein glycosylation have been reported in various diseases, including cancer; however, the consequences of altered glycosylation in meningiomas remains undefined. We established two benign meningioma cell lines—SUT-MG12 and SUT-MG14, WHO grade I—and demonstrated the glycan and glycosyltransferase profiles of the mucin-type O-linked glycosylation in the primary benign meningioma cells compared with two malignant meningioma cell lines—HKBMM and IOMM-Lee, WHO grade III. Changes in O-linked glycosylation profiles in malignant meningiomas were proposed.

Methods

Primary culture technique, morphological analysis, and immunocytochemistry were used to establish and characterize two benign meningioma cell lines. The glycan profiles of the primary benign and malignant meningiomas cell lines were then analyzed using lectin cytochemistry. The gene expression of O-linked glycosyltransferases, mucins, sialyltransferases, and fucosyltransferases were analyzed in benign and malignant meningioma using the GEO database (GEO series GSE16581) and quantitative-PCR (qPCR).

Results

Lectin cytochemistry revealed that the terminal galactose (Gal) and N-acetyl galactosamine (GalNAc) were highly expressed in primary benign meningioma cells (WHO grade I) compared to malignant meningioma cell lines (WHO grade III). The expression profile of mucin types O-glycosyltransferases in meningiomas were observed through the GEO database and gene expression experiment in meningioma cell lines. In the GEO database, C1GALT1-specific chaperone (COSMC) and mucin 1 (MUC1) were significantly increased in malignant meningiomas (Grade II and III) compared with benign meningiomas (Grade I). Meanwhile, in the cell lines, Core 2 β1,6-N-acetylglucosaminyltransferase-2 (C2GNT2) was highly expressed in malignant meningiomas. We then investigated the complex mucin-type O-glycans structures by determination of sialyltransferases and fucosyltransferases. We found ST3 β-galactoside α-2,3-sialyltransferase 4 (ST3GAL4) was significantly decreased in the GEO database, while ST3GAL1, ST3GAL3, α1,3 fucosyltransferases 1 and 8 (FUT1 and FUT8) were highly expressed in malignant meningioma cell lines—(HKBMM)—compared to primary benign meningioma cells—(SUT-MG12 and SUT-MG14).

Conclusion

Our findings are the first to demonstrate the potential glycosylation changes in the O-linked glycans of malignant meningiomas compared with benign meningiomas, which may play an essential role in the progression, tumorigenesis, and malignancy of meningiomas.

Introduction

Meningiomas are the most common intracranial tumors arising from the arachnoid space of the brain. The WHO 2021 classifies meningiomas according to molecular markers and histological characteristics (viz., mitotic activity, brain invasion, and presence of other minor criteria) as benign (grade I), atypical (grade II), or anaplastic (grade III) (Goldbrunner et al., 2016; Maier et al., 2020; Li & Drappatz, 2023; Louis et al., 2021). Meningioma grade I is the most frequent subtype (>80%), with slow growth and low risk of recurrence, while remaining 20% for grade II and grade III are shown aggressive biological behavior and high recurrence (Li & Drappatz, 2023; Champeaux et al., 2016). Meningiomas are classified as grade II in the presence of at least four to nine mitotic figures in ten consecutive high-power field (10 HPF) and/or brain invasion whereas meningioma grade III (anaplastic, rhabdoid, and papillary subtype) have 20 or more mitotic figures in 10 HPF or frank histological anaplasia with a morphology resembling a carcinoma, melanoma, or sarcoma, and/or TERT promoter (pTERT) mutation, and/or CDKN2A/B homozygous deletion (Louis et al., 2021; Champeaux et al., 2016). Meningiomas grade III are rare (1–3%), so details of their malignancy are limited. Surgical, chemical, and radiotherapeutic treatments are still being developed to manage the morbidity and mortality of grade III meningiomas (Champeaux et al., 2016; Buerki et al., 2018; Peyre et al., 2018).

O-linked glycosylation is a post-translational modification in which the glycans (oligosaccharides) are attached to a serine or threonine residue of polypeptide chains (proteins). Mucin-type O-linked glycosylation is found in abundance on mucins, which carry hundreds of glycan structures in this linkage (Zhang et al., 2022). Glycosylated proteins are essential in cell functions, host-pathogen interactions, inflammation, development, and malignancy (Matsumoto & Ju, 2023). Changes in protein glycosylation can result in abnormal structure and function of proteins or enzyme activities (Caval, Alisson-Silva & Schwarz, 2023; Gautam et al., 2023). Investigation of altered cancer-related glycoprotein expression may yield the discovery of potential biomarkers and novel targets for therapeutics.

Glycosyltransferases are key enzymes in the glycosylation changes of cancers (Pucci et al., 2022). Reports of tumor-associated O-linked glycosylation includes Core 1 β1-3 galactosyltransferase (C1GALT1), C1GALT1-specific chaperone (COSMC), Core 2 β1,6-N-acetylglucosaminyltransferase-1 (C2GNT1), Core 2 β1,6-N-acetylglucosaminyltransferase-2 (C2GNT2), and Core 3 β1,3-N-acetylglucosaminyltransferase 6 (B3GNT6) (Curigliano et al., 2020; Martinez-Saez, Peregrina & Corzana, 2017; Miyamoto et al., 2013; Okamoto et al., 2013; Schachter & Brockhausen, 1989; Sindrewicz, Lian & Yu, 2016). Down-regulation of C1GALT1 and its chaperone, COSMC, contributed to increased truncation of O-glycans, Tn and sialyl-Tn (sTn) antigens, on mucin glycoproteins (Chugh et al., 2018; Khiaowichit et al., 2022; Liu et al., 2020; Sagar et al., 2021). C2GNT1 competes with ST3 β-galactoside α-2,3-sialyltransferase 1 (ST3GAL1) and further truncated O-glycosylation by capping the T-antigen with sialic acid (Chandrasekaran et al., 2006). Overexpression of C2GNT1 results in altered O-glycosylation of prostate-specific antigen (PSA), prostatic acidic phosphatase (PAP) and mucin 1 (MUC1) in prostate cancer cells (Chen et al., 2014). Modified MUC1, due to overexpression of C2GNT, results in immune evasion of bladder cancer (Suzuki et al., 2012). The roles of B3GNT6-associated tumor progression have been demonstrated in pancreatic, prostatic, colon, and colorectal cancer (Gupta et al., 2020). The extension of O-glycans is known to form structures such as Sialyl-Lewis X (sLex) and Sialyl-Lewis A (sLea) by capping with sialyltransferases (STs) and fucosyltransferases (FUTs). These O-linked glycan complex structures were involved in cancer cell invasion and metastasis (Thomas, Rathinavel & Radhakrishnan, 2021).

In the current study, we established the primary benign meningioma cells (WHO grade I) and compared the expression of glycans, mucins, glycosyltransferases, sialyltransferases, and fucosyltransferases with the malignant meningioma cell lines (WHO grade III). Changes in O-linked glycosylation in malignant meningiomas were proposed. Knowledge of these specific O-linked glycosylations may help to understand the mechanisms of tumorigenesis, progression, and metastasis of meningiomas. There may, also be applications for tumor prognostic or/and specific treatment.

Materials and Methods

Human malignant meningioma cell lines and culture

IOMM-Lee, a human meningioma cell line (WHO grade III), was purchased from the American Type Culture Collection (Manassas, VA, USA). HKBMM, a human meningioma cell line (WHO grade III), was kindly provided by Associate Professor Norie ARAKI (Kumamoto University, Kumamoto, Japan). Both cell lines were cultured in complete medium containing Dulbecco’s modified Eagle’s medium (DMEM, cat. no. 12100-046; Thermo Fisher Scientific, Inc., Waltham, MA, USA) supplemented with 1% penicillin-streptomycin (cat. no. 15140-122; Thermo Fisher Scientific, Inc., Waltham, MA, USA) and 10% fetal bovine serum (FBS, cat. no. 10270-098; Thermo Fisher Scientific, Inc., Waltham, MA, USA) and maintained in a humidified incubator with 5% CO2 at 37 °C.

Human benign meningioma cells isolation and culture

Benign meningioma tissues were obtained from two patients who underwent surgical resection at Suranaree University of Technology Hospital, Thailand and were kept in a frozen medium containing 90% FBS, and 10% DMSO. Written informed consent was obtained from each subject. The Suranaree University of Technology Ethics Committee approved the study protocol (registration number: EC-64-154). The two benign meningioma tissues were designated as SUT-MG12 and SUT-MG14. SUT-MG12 and SUT-MG14 were derived from a 70- and 63-year-old female patient, respectively. The histopathological examination of the two benign meningioma tissues was determined and confirmed to be meningothelial meningioma, WHO grade I. The frozen benign meningioma tissues were rinsed in PBS and resuspended in a complete medium for primary cell establishment. The tissues were minced into 2–3 mm3 pieces. Tissue fragments were transferred into a 15 mL tube for the washing step. The fragments were resuspended in pre-warmed complete medium with collagenase solution at 1 mg/mL (cat no. 17100-017; Thermo Fisher Scientific, Inc., Waltham, MA, USA) and incubated for 45 min at 37 °C with gentle stirring every 5 min. The digested tissues were sieved through 100 and 70 µm nylon filters (cat nos. 93070 and 93100; SPL Life Sciences, Pochon, South Korea). Cell suspensions were centrifuged and washed with complete medium. The cells were then resuspended in DMEM with 1% penicillin-streptomycin, 20% fetal bovine serum, and 2 mM L-glutamine (cat. no. 25030-081; Thermo Fisher Scientific, Inc., Waltham, MA, USA) and maintained in a humidified incubator with 5% CO2 at 37 °C. Primary benign meningioma cells with 90% confluence were split by trypsinization. The medium was changed twice a week. The passage number of the primary cell culture was between 3 and 10, after applying subsequent experiments.

Immunocytochemistry and lectin staining

Cells were seeded at 3–5 × 105 cells into a 24-well plate. After seeding for 48 h, cells were fixed with 4% paraformaldehyde in PBS for 15 min and permeabilized with 0.2% Triton X-100 in PBST for 10 min. Endogenous hydrogen peroxide activity was blocked with 0.3% of hydrogen peroxide for 30 min. Then, nonspecific binding was blocked with 3% bovine serum albumin (BSA) for 30 min. Sixteen biotinylated lectins (Vector Laboratories, Inc., Newark, CA, USA) or primary antibodies were probed at 25 °C overnight, on a shaker including (cat.no. BK-1000 and BK-3000; Vector Laboratories, Inc., Newark, CA, USA), vimentin (dilution 1:100, cat. no. SC-6260; Santa Cruz Biotechnology, Inc., Dallas, TX, USA), MUC1 (dilution at 1:150 cat. no. SC-7313; Santa Cruz Biotechnology, Inc., Dallas, TX, USA), and SSTR2A (dilution at 1:100, cat. no. Ab134152; Abcam, Cambridge, UK) (Table 1). Then, ABC-Peroxidase Solution (cat. no. PK-4000; Vector Laboratories, Inc., Newark, CA, USA) was used for 1 h at room temperature as the secondary antibody to determine the lectin signal. Meanwhile, horseradish peroxide-conjugated secondary antibody (1:100), (GE Healthcare, Buckinghamshire, UK) was used for 1 h to determine the protein expression of vimentin, MUC-1, and SSTR2A. Visualization was accomplished using a SignalStain® DAB substrate kit (cat. No. 8059; Cell Signaling Technology, Inc., Danvers, MA, USA). Images were visualized using an inverted microscope with an original magnification of ×200.

Table 1 Lectin cytochemistry of primary benign and malignant meningiomas cells.

Lectin	Major sugar specific	SUT-MG12	SUT-MG14	IOMM-lee	HKBMM	
ConA	Mannose	+++	+++	+++	+++	
UEAI	Fucose	+	+	–	+	
DBA	N-acetylgalactosamine	++	++	+	++	
VVL	N-acetylgalactosamine	++	++	–	+	
RCA120	N-acetylgalactosamine	+++	+++	++	+++	
SBA	N-acetylgalactosamine	+	+	–	+	
PNA	Galactose	++	++	–	–	
ECL	Galactose	+++	+++	–	+	
Jacalin	Galactose	+	+	–	–	
WGA	N-acetylglucosamine	+++	+++	+++	+++	
GSLII	N-acetylglucosamine	++	++	+	+	
DSL	N-acetylglucosamine	++	++	+	+	
LEL	N-acetylglucosamine	+	–	+	+	
STL	N-acetylglucosamine	+	+	–	–	
MALII	(α-2,3) linked sialic acid	+	+	–	++	
SNA	(α-2,6) linked sialic acid	++	++	–	++	
Note:

Con A, Concanavalin A; UEA I, Ulex europaeus agglutinin I; DBA, Dolichos biflorus agglutinin; VVL, Vicia villosa agglutinin; RCA120, Ricinus communis agglutinin; SBA, Glycine max (soybean); PNA, Arachis hypogaea (peanut) agglutinin; ECL, Erythrina cristagalli lectin; Jacalin; WGA, Triticum vulgaris (wheat germ); GSL II, Griffonia (Bandeiraea) simplicifolia lectin II; DSL, Datura Stramonium lectin; LEL, Lycopersicon esculentum (tomato) lectin; STL, Solanum tuberosum (potatoe) lectin; MAL II, Maackia Amurensis Lectin II (MAL II); SNA, Sambucus nigra lectin.

Gene expression analysis by GEO database

The gene expression data (GEO series GSE16581) of the meningiomas were retrieved from the Gene Expression Omnibus (GEO) database ((https://www.ncbi.nlm.nih.gov) accessed on 20 June 2022). The GEO series GSE16581 comprises expression data from 43 WHO grade I, 19 WHO grade II and 6 WHO grade III meningiomas. All expression data were log2 transformed.

Gene expression analysis by quantitative PCR

Cells with 90% confluence in a 60 mm culture dish were harvested by TRIzol reagent (cat. no. 15596026; Thermo Fisher Scientific, Inc., Waltham, MA, USA). Total RNA was extracted according to the manufacturer’s protocol. The cDNA synthesis using the SensiFAST cDNA synthesizer kit was performed. The primer sets for all the glycosyltransferases are listed in Table 2. The mRNA expression levels of all the glycosyltransferases were analyzed in both primary benign meningioma cells and malignant meningioma cell lines using quantitative PCR (qPCR) as previously described (Talabnin, Talabnin & Wongkham, 2020). The gene amplification condition was performed as previously described (Talabnin, Talabnin & Wongkham, 2020). β-Actin was used as the internal control to normalize the expression of the target genes. Relative mRNA expression was determined using the 2Ct method (Livak & Schmittgen, 2001).

Table 2 Sequences of the primers used for reverse transcription-quantitative PCR.

Gene	Forward (5′-3′)	Reverse (5′-3′)	
O-liked glycosyltransferases	
C1GALT1	AAGCAGGGCTACATGAGTGG	GCATCTCCCCAGYGCTAAGT	
COSMC	AAC GTG AGA GGA AAC CCG T	AAA GCA TTT TTC CCG CGT C	
C2GNT1	GACGTTGCTGCGAAGG	CCAAGTGTCTGACACTTACA	
C2GNT2	GCGAAAGAACCCTCAATCAG	GCTGCAGTTTCCCTTCAGTC	
B3GNT6	TCA ACC TCA CGC TCA AGC AC	CAG GAA GCG GAC TAC GTT GG	
Mucin	
MUC1	TTTCCAGCCCGGGATACCTA	TAGGGGCTACGATCGGTACT	
MUC3	TGGCTGAGCAACAACTCTGT	GGGAGGCTACTGTTGGTTT	
MUC4	TCACCTCAACAGGCTCAACA	GTCATCATCTGCGTGAGGGT	
MUC13	TAATCACCGCTTCATCTCCA	TGTTTAGGGTGCTGGTCTCC	
Sialyltransferases	
ST3GAl I	GGACCCTGAAAGTGCTCA	TCTCCAGCATAGGGTCCA	
ST3Gal III	GTATGATCGGTTGGGCTT	CGCTCGTACTGCTCAGG	
ST3Gal IV	GTCAGCAAGTCCAGCT	CTTGTTGATGGCATCTCCC	
ST3Gal VI	GGTATCTTGTGGCCATATTCC	CTCCATTACCAACCACCAC	
ST6GalNAc1	TATCGTAAGCTGCACCCCAATA	TTAGCAGTGAATGGTCCGGAAA	
ST6GAL1			
Fucosyltransferase	
FUT1	TGAGGGATCACTGCCAAAATG	TCTTGGCAGTTTATGAGCTTTAAAAA	
FUT2	GCTCGCTACAGCTCCCTCAT	CGTGGGAGGTGTCAATGTTCT	
FUT3	GCCGACCGCAAGGTGTAC	TGACTTAGGGTTGGACATGATATCC	
FUT4	AAGCCGTTGAGGCGGTTT	ACAGTTGTGTATGAGATTTGGAAGCT	
FUT5	TATGGCAGTGGAACCTGTCA	CGTCCACAGCAGGATCAGTA	
FUT6	CAAAGCCACATCGCATTGAA	ATCCCCGTTGCAGAACCA	
FUT7	TCCGCGTGCGACTGTTC	GTGTGGGTAGCGGTCACAGA	
FUT8	TTGCCATTTATGCTCACCAA	TTCCAGCCACACCAATGATA	
FUT9	TCCCATGCAGTTCTGATCCAT	GAAGGGTGGCCTAGCTTGCT	
ACTB	GAT CAG CAA GCA GGA GTA TGA CG	AAG GGT GTA ACG CAA CTA AGT CAT AG	
Note:

C1GALT1, Core 1 β1-3 Galactosyltransferase; COSMC, C1GALT1-specific chaperone; C2GnT1, Core 2 β1,6-N-acetylglucosaminyltransferase-1; C2GnT2, Core 2 β1,6-N-acetylglucosaminyltransferase-2; B3GNT6, Core 3 β1,3-N-acetylglucosaminyltransferase 6; MUC1, 3, 4, 13, Mucin 1, 3, 4, 13; ST3Gal I, III, IV, VI, ST3 β-Galactoside α-2,3-Sialyltransferase 1, 3, 4, 6; ST6GalNAc-I, ST6 N-Acetylgalactosaminide α-2,6-Sialyltransferase 1; ST6GAL1, ST6 β-galactoside α-2,6-sialyltransferase 1; FUT1-9, α1,3 fucosyltransferases 1-9; ACTB, β-actin.

Statistical analysis

The independent t-test was used to determine the differential expression of glycosyltransferase genes between the primary benign meningiomas and the malignant meningiomas. All analyses used GraphPad Prism software (version 8.0; GraphPad Software, Inc., La Jolla, CA, USA). A P < 0.05 indicated a statistically significant difference.

Results

Establishment of the primary benign meningioma cells

A well-characterized benign meningioma cell line is not available due to the senescence of nonmalignant cells occurring in vitro. Thus, two primary benign meningioma cells (WHO grade I)—SUT-MG12 and SUT-MG14—were isolated from two meningothelial meningioma tissues and performed primary culture technique. The morphological analysis demonstrated that SUT-MG12 and SUT-MG14 are spindle-shaped cells with large nuclei (Fig. 1). The primary culture showed that SUT-MG12 and SUT-MG14 could proliferate and multiply after more than 10 passages, but the proliferation rate declined after passage no. 10, and the cells were enlarged and underwent senescence. Thus, the cells in passage nos. 3–10 were used for subsequent experiments. Using immunocytochemical analysis, protein expressions of vimentin (Vim) and somatostatin receptor 2A (SSTR2A) were observed in these two primary benign meningioma cells (WHO grade I) including the well-established meningioma cell lines—HKBMM and IOMM-Lee (WHO grade III). However, the protein expression of mucin 1 (MUC1) was nominal (Fig. 1). The data from the primary culture, morphological analysis, and immunocytochemistry demonstrated the benign meningiomas characteristic of SUT-MG12 and SUT-MG14.

Figure 1 Characterization of primary benign meningioma cells.

Cell morphology by hematoxylin and eosin staining (H&E). Immunocytochemistry of somatostatin receptor 2A (SSTR2A), mucin and vimentin in primary benign and malignant meningiomas cells. (Magnification, 200×).

Alteration of glycan expression in meningiomas

Altered expression of glycans have been associated with tumor development and progression (Thomas, Rathinavel & Radhakrishnan, 2021). For screening of the potential glycan structures between the benign and malignant meningiomas, a specific lectin technique is a good candidate. Using lectin-cytochemistry, the glycan profiles of primary benign meningioma cell lines—SUT-MG12 and SUT-MG14—were compared with the malignant meningioma cell lines—HKBMM and IOMM-Lee. Sixteen lectins with different glycan specificities (Table 1) were investigated. ConA (for mannose), RCA (for GalNAc), WGA, and GSLII (for GlcNAc) were stained strongly positive in both primary benign and malignant meningioma cell lines. At the same time, UEAI (for fucose), SBA (for GalNAc), DSL, LEL, and STL (for GlcNAc) were only slightly stained (positive). In contrast, DBA, VVL (for GalNAc), PNA, ECL, and Jacalin (for Gal) were strongly stained (positive) in primary benign meningioma cell lines compared with negative or slightly positive stained in malignant meningioma cell lines (Figs. 2–4). Furthermore, the alteration of sialylation in meningiomas was determined using MAL II (alpha 2,3 sialylation) and SNA (alpha 2,6 sialylation). Strong positive staining of SNA was observed in primary benign meningioma cell lines (Fig. 4, Table 1). These findings demonstrated the differential expression of glycans in the primary benign and malignant meningiomas, the high expression of GalNAc, Gal glycans, and alpha 2,6 sialylation was observed in primary benign meningioma cell lines—SUT-MG12 and SUT-MG14 (WHO grade I)—compared with malignant meningioma cell lines—HKBMM and IOMM-Lee (WHO grade III).

Figure 2 Lectin cytochemistry in primary benign and malignant meningiomas.

Lectins for mannose (ConA), fucose (UEA I), and N-acetyl galactosamine (DBA, VVL, RCA120, SBA), (Magnification, 200×).

Figure 3 Lectin cytochemistry in primary benign and malignant meningiomas.

Lectins for N-acetyl glucosamine (WGA, GSL-II, DSL, LEL, STL), (Magnification, 200×).

Figure 4 Lectin cytochemistry in primary benign and malignant meningiomas.

Lectins for galactose (PNA, ECL, Jacalin) and sialic acid (MAL II, SNA), (Magnification, 200×).

Altered expression of mucin-type O-linked glycosyltransferases in meningiomas

To explore whether the specific changes of the O-linked glycosylation in benign and malignant meningiomas are due to the aberrant expression of the mucin-type O-linked glycosyltransferase. The expression of the mucin-type O-linked glycosyltransferase genes—including C1GALT1, COSMC, C2GNT1, C2GNT2, and B3GNT6—were investigated through the GEO database (GSE16581) and all underwent qPCR. In the GEO database, the respective mRNA expressions of C1GALT1, COSMC, C2GNT1, C2GNT2, and B3GNT6 was highly expressed in Grade I, Grade II, and Grade III meningiomas (Fig. 5A). Meanwhile, COSMC was only significantly increased in Grade II and III vs Grade I meningiomas (P < 0.05). We then further investigated the mucin-type O-linked glycosyltransferase gene using qPCR in primary benign—SUT-MG12 and SUT-MG14—and malignant—HKBMM and IOMM-Lee—meningioma cell lines. The gene expression experiments demonstrated that mRNA expression of C1GALT1 and COSMC were highly expressed in both the primary benign and malignant meningioma cell lines. In contrast, C2GNT2 was highly expressed in malignant meningioma cell lines compared to primary benign meningioma cells (Fig. 5B).

Figure 5 Mucin-type O-linked Glycosyltransferases expressions in benign and malignant meningiomas.

(A) The mRNA expression levels of Mucin-type O-linked glycosyltransferases in meningioma were obtained from the GEO database (GEO series GSE16581) and *P < 0.05 vs WHO Grade I. (B) The relative mRNA expression of mucin-type O-linked glycosyltransferases were determined using qPCR. Expression values were presented as mean + SEM of three independent experiments.

Additionally, investigations of membrane-bound mucin genes (MUC1, MUC3, MUC4, and MUC13), associated with altered O-linked glycosylation in various cancers, were investigated through the GEO database (GSE16581) and underwent qPCR. In the GEO database, the respective mRNA expressions of MUC1, MUC3, MUC4, and MUC13 was detected in Grade I, Grade II, and Grade III meningiomas. Still, only MUC1 expression was significantly increased in grade II and III meningiomas (Fig. 6A). In contrast, MUC1 and MUC4 were highly expressed in primary benign—SUT-MG12 and SUT-MG14—meningioma cell lines (Fig. 6B).

Figure 6 Membrane-bound mucin expressions in benign and malignant meningiomas.

(A) The mRNA expression levels of membrane-bound MUCINs in meningioma were obtained from the GEO database (GEO series GSE16581) and *P < 0.05 vs WHO grade I. (B) The relative mRNA expression of membrane-bound MUCINs were determined using qPCR. Expression values were presented as mean + SEM of three independent experiments.

Altered sialyltransferases expression in meningiomas

Changes in sialylated glycans have been linked to facilitate malignant cell transformation, immune evasion, and metastatic spread (Hugonnet et al., 2021). Additionally, high expression of alpha 2,6 sialylation was observed in primary benign meningioma cell compared to malignant meningioma cell lines (Fig. 4). To draw a full picture of the alteration of O-linked glycosylation in benign and malignant meningiomas, the expression of “capping enzymes”—sialyltransferases were demonstrated. For the extension of the mucin-type O-linked glycosylation, the expression of six sialyltransferases (STs)—ST3GAL1, ST3GAL3, ST3GAL4, ST3GAL6, ST6GAL1, and ST6GALNAc1—were investigated using the GEO database (GSE16581) and underwent qPCR. In the GEO database, mRNA expression of ST3GAL4 was significantly decreased in grade II and III meningiomas (P < 0.05), while the others (ST3GAL1, ST3GAL3, ST3GAL6, ST6GAL1, and ST6GALNAc1) were not (Fig. 7A). In the meningioma cell lines, mRNA expression of ST6GAL1 was highly expressed in the primary benign meningiomas—SUT-MG12 and SUT-MG14—while the mRNA expression of ST3GAL1 and ST3GAL3 was observed in the malignant meningioma cell lines—HKBMM and IOMM-Lee (Fig. 7B).

Figure 7 Sialyltransferase (STs) expressions in benign and malignant meningiomas.

(A) The mRNA expression levels of sialyltransferase in meningioma were obtained from the GEO database (GEO series GSE16581) and *P < 0.05 vs WHO Grade I. (B) The relative mRNA expression of sialyltransferase were determined using qPCR. Expression values were presented as mean + SEM of three independent experiments.

Altered fucosyltransferase expression in meningiomas

Lewis antigens are terminal fucosylated carbohydrate motifs decorating cell surface N- and O-linked glycoproteins by capping with fucosyltransferases (FUTs). Additionally, overexpression of different FUTs during malignant cell transformation are associated with the acquisition of an increased proliferative capacity and a pro-survival phenotype (Lv et al., 2023). To address the complex structures of the O-linked glycosylation in benign and malignant meningiomas, the expressions of fucosyltransferases (FUTs) were determined. Fucosylation of the mucin-type O-linked glycosylation may generate the complex structure, i.e., Sialyl-Lewis X (sLex) and Sialyl-Lewis A (sLea) in meningiomas. Thus, the expression of nine fucosyltransferases (FUTs) were investigated (viz., FUT1, FUT2, FUT3, FUT4, FUT5, FUT6, FUT7, FUT8, and FUT9). In the GEO database, there was no respective significantly different expression of FUT1, FUT2, FUT3, FUT4, FUT5, FUT6, FUT7, FUT8, or FUT9 in grade I vs grade II & III meningiomas (Fig. 8A). However, the gene expression experiments demonstrated that FUT1 and FUT8 were highly expressed in the malignant meningioma—HKBMM—vs the primary benign meningiomas—SUT-MG12 and SUT-MG14 (Fig. 8B). Due to the high expression of FUT1 and FUT8 in HKBMM cell lines, we removed FUT1 and FUT8 results from the analysis. We found that the respective expression of FUT2, FUT3, FUT6, and FUT7 was also increased in the malignant meningiomas—HKBMM and IOMM-Lee (Fig. 8C).

Figure 8 Fucosyltransferase (FUTs) expressions in benign and malignant meningiomas.

(A) The mRNA expression levels of fucosyltransferase in meningioma were obtained from the GEO database (GEO series GSE16581). (B and C) The relative mRNA expression of fucosyltransferase were determined using qPCR. Expression values were presented as mean + SEM of three independent experiments.

Discussion

Understanding the biological changes of tumor cells is essential for finding diagnostics, prognostics, therapeutics, and/or preventions of cancers. In the current study, we determined the glycans and glycosyltransferase profiles between the benign and malignant meningiomas. Firstly, we established two benign meningioma cell lines (SUT-MG12 and SUT-MG14, WHO grade I) and delineated their characteristics compared to two malignant meningioma cell lines (HKBMM and IOMM-Lee, WHO grade III). Secondly, the glycan profiles and the differential mRNA expression of the glycosyltransferases, mucins, sialyltransferases, and fucosyltransferases of meningiomas were investigated in silico—GEO database, GSE16581—and in vitro—primary benign and malignant meningioma cell lines. Finally, the O-linked glycosylation biosynthetic pathways of malignant meningiomas were proposed (Fig. 9).

Figure 9 Proposed structures of the mucin type O-linked glycosylation in malignant meningiomas (WHO grade III).

High expression of C1GALT1, COSMC, C2GNT2, ST3GAL1, ST3GAL3, FUT1, FUT2, FUT3, FUT6, FUT7, and FUT8 were demonstrated in malignant meningiomas.

Due to the lack of in vitro meningiomas models, the pathogenesis of this tumor remains unclear (Mei et al., 2017). Herein, we established two primary benign meningioma cell lines (WHO grade I)—SUT-MG12 and SUT-MG14—using the primary culture technique, morphological analysis, and immunocytochemistry. SUT-MG12 and SUT-MG14 were confirmed as benign meningioma cells by comparing them with the well-established malignant meningioma cell lines (viz., HKBMM and IOMM-Lee) (Mei et al., 2017; Püttmann et al., 2005). Please note that, WHO grade II cell line was not investigated in this present study due to the limitation of clinical meningioma cases.

Lectin-cytochemistry demonstrated different expressions of the glycan profiles in the primary benign and malignant meningioma cell lines. High expression of Gal and GalNAc in primary benign meningioma cell lines (viz., SUT-MG12 and SUT-MG14) may represent high expression of the common core structures—the Tn (GalNAc1) and T (Gal1GalNAc1 or Core 1) antigens. These Tn and T antigens—known as pan-carcinoma antigens—have been reported across many types of cancers (i.e., breast, colon, gastric, pancreatic, respiratory, and melanoma) (Kudelka et al., 2015). This may be due to the aberrant expression of polypeptide GalNAc-transferases which transfer a GalNAc moiety from UDP-GalNAc onto Ser or Thr residues of the O-linked glycoproteins. Thus, further studies on the expression of polypeptide GalNAc-transferase are required (Thomas, Rathinavel & Radhakrishnan, 2021). However, in the present study, the respective expression of Gal and GalNAc was decreased in the malignant meningioma cell lines—HKBMM and IOMM-Lee. Taken together, changes in glycan expression in the benign and malignant meningiomas suggest their role vis-à-vis the cellular characteristics of this tumor. Notably, the structural details of the glycans and their quantities should be further investigated by quantitative techniques such as mass spectrometry or high-performance liquid chromatography.

Altered expression of the mucin-type O-linked glycosyltransferases in meningiomas was demonstrated through the GEO database (GSE16581) and gene expression experiments in the meningioma cell lines. In the GEO database (GSE16581), COSMC was significantly increased in malignant meningioma (Grade II and III) compared with benign meningioma (grade I). COSMC helps C1GALT1 to fold correctly and maintain activity. It also involves in development of immune-mediated disease, inflammation, and cancer (Xiang et al., 2022). Specifically, C1GALT1 and COSMC were highly expressed in both the primary benign and malignant meningioma cell lines, suggesting a high expression of core 1 (T antigen, Gal1GalNAc1) in meningiomas. In contrast, C2GNT2—which is responsible for the core 2, core 4, and branched polylactosamine structures (I-branches)—were highly expressed in the malignant meningioma cell lines (HKBMM and IOMM-Lee) compared with the benign meningioma cell lines (SUT-MG12 and SUT-MG14). These results are consistent with the lectin-cytochemistry in which high expression of Gal and GalNAc in the primary benign meningioma cell lines. Additionally, the elongation of O-linked glycans may increase in the malignant meningioma cell lines. These finding suggest that C2GNT2 and its glycan structures—core 2, core 4, and branched polylactosamines—may be involved in the invasion and recurrence of malignant meningiomas (Selke et al., 2021).

Expression of mucins, particularly mucin 4 (MUC4), is widely expressed in meningiomas (Matsuyama et al., 2019). In silico, MUC1, MUC3, MUC4, and MUC13 were highly expressed in meningiomas (GEO database series GSE16581). In vitro, MUC1 and MUC4 were highly expressed in the benign meningioma cell lines—SUT-MG12 and SUT-MG14—, but only MUC1 was highly expressed in the malignant meningioma cell lines—HKBMM and IOMM-Lee. These data suggest that mucin 1 and mucin 4 glycoconjugates play crucial roles in benign meningiomas, while in malignant meningiomas, the mucin 1 glycoconjugates are the major contributors. Additionally, several studies demonstrated that upregulated MUC1 and MUC4 expression plays an important role in tumor cell metabolism, mesenchymal transition, apoptosis, and metastasis (Qing, Li & Dong, 2022). However, the exact mechanism of MUC1 or MUC4 in the development and progression of meningioma is still unclear. Thus, further investigations on the role of MUC1 and MUC4 are required.

For the extension of mucin-type O-linked glycosylation, sialylation and fucosylation were investigated. ST6GAL1 was highly expressed in the primary benign meningioma cells—SUT-MG12 and SUT-MG14. This gene encodes the protein that catalyzes the transfer of sialic acid from CMP-sialic acid to galactose-containing substrates. ST6GAL1 inhibits metastasis of hepatocellular carcinoma via modulating the sialylation of MCAM on the cell surface (Garnham et al., 2019). High expression of ST6GAL1 in benign meningiomas may be involved in regulating benign tumor characteristics.

In contrast, ST3GAL1 and ST3GAL3 were highly expressed in the malignant meningioma—HKBMM and IOMM-Lee. ST3GAL1 is a major sialyltransferase in human to synthesized sialyl-T (ST) antigen (NeuAc2,3-Galβ1-3GalNAc) from T antigen (Galβ1-3GalNAc). At the same time, ST3GAL3 is responsible for the synthesis of Sialyl-Lewis A (sLea) and Sialyl-Lewis X (sLex). Aberrant expression of ST3GAL1 and ST3GAL3 was found in different cancer types and has been associated poor prognosis and cancer progression (Hugonnet et al., 2021). High expression of ST3GAL1 and ST3GAL3 in malignant meningiomas may play an essential role in the progression and invasion of this malignancy.

Fucosylation of the mucin-type O-linked glycosylation may generate complex structures in meningiomas—i.e., Sialyl-Lewis X (sLex) and Sialyl-Lewis A (sLea) antigens. FUT1, FUT2, FUT3, FUT6, FUT7, and FUT8 were highly increased in malignant meningioma cell lines. The aberrant expression of FUTs occurs in many cancer cells. Recent studies revealed that increased FUTs are a signature of malignant cell transformation and contribute to many abnormal events during cancer development, such as uncontrolled cell proliferation, tumor cell invasion, angiogenesis, metastasis, immune evasion, and therapy resistance (Lv et al., 2023). High expression of FUT1, FUT2, FUT3, FUT6, FUT7, and FUT8 in malignant meningioma cell lines—HKBMM and IOMM-Lee—may represent a fucosylation signature of mucin-type O-linked glycosylation in malignant meningiomas.

The limitations of the present study still need to be addressed. Firstly, meningioma WHO grade II cell line was not investigated in this present study due to the limitation of clinical meningioma cases. Thus, the verification of the results in meningioma WHO grade II cell line is required in further investigation. Secondly, several candidate glycans and glycosyltransferase genes including mucin-type O-linked glycosyltransferases (C2GNT2), sialyltransferases (ST3GAL1, ST3GAL3, and ST6GAL1), and fucosyltransferases (FUT1, and FUT8) were observed in the present study. However, further studies on the exact glycans and their quantities should be performed by quantitative techniques such as mass spectrometry or high-performance liquid chromatography. Moreover, the verification of the results with large sample size, protein expression experiments, and in vitro experiments of the biological effect of each candidate gene are required in further investigation to provide the significant role of each candidate gene on the development and progression of meningioma.

In conclusion, the current study confirmed intrinsic biological pathways of the O-linked glycosylation in benign and malignant meningiomas. Alteration of specific glycans and glycosyltransferases may reflect their role in tumor progression. Increased expression of C2GNT2, ST3GAL1, ST3GAL3, FUT1, FUT2, FUT3, FUT6, FUT7, and FUT8 in malignant meningiomas, which generate the complex glycan structures, may play an essential role in invasion and recurrence of malignant meningiomas, which could be further studied for their prognostic or therapeutic applications.

Supplemental Information

Supplemental Information 1 MIQE checklist.

Click here for additional data file.

Supplemental Information 2 Raw data for Figure 3A.

Mucin type O-glycosyltransferase.

Click here for additional data file.

Supplemental Information 3 Raw data for Figure 4A.

Membrane-bound mucins.

Click here for additional data file.

Supplemental Information 4 Raw data for Figure 5A.

Sialyltransferase Grade I vs Grade II & III.

Click here for additional data file.

Supplemental Information 5 Raw data for Figure 6A.

Fucosyltransferase Grade I vs Grade II & III.

Click here for additional data file.

The authors thank (a) the patients and their families for their participation; (b) Suranaree University Hospital for providing patient samples; and (c) Mr. Bryan Roderick Hamman for assistance with the English-language presentation of the present study.

Additional Information and Declarations

Competing Interests

Author Contributions

Human Ethics

Data Availability

The authors have declared that no competing interest exists.

Chutima Talabnin conceived and designed the experiments, performed the experiments, analyzed the data, prepared figures and/or tables, authored or reviewed drafts of the article, and approved the final draft.

Thanawat Trasaktaweesakul performed the experiments, analyzed the data, prepared figures and/or tables, and approved the final draft.

Pitchanun Jaturutthaweechot performed the experiments, analyzed the data, prepared figures and/or tables, and approved the final draft.

Pundit Asavaritikrai performed the experiments, analyzed the data, prepared figures and/or tables, and approved the final draft.

Dusit Kongnawakun performed the experiments, analyzed the data, prepared figures and/or tables, and approved the final draft.

Atit Silsirivanit conceived and designed the experiments, analyzed the data, prepared figures and/or tables, and approved the final draft.

Norie Araki conceived and designed the experiments, analyzed the data, prepared figures and/or tables, and approved the final draft.

Krajang Talabnin conceived and designed the experiments, performed the experiments, analyzed the data, prepared figures and/or tables, authored or reviewed drafts of the article, and approved the final draft.

The following information was supplied relating to ethical approvals (i.e., approving body and any reference numbers):

The Ethics Committee of Suranaree University of Technology approved the study protocol (registration number: EC-64-154).

The following information was supplied regarding data availability:

The raw data of Figures 5, 6, 7, and 8 are available in the Supplemental Files.

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
