# Peer review of "Altered O-linked glycosylation in benign and malignant meningiomas"

_PeerJ, doi:10.7717/peerj.16785_

## Round 0.1 · original submission · Major Revisions

As I see the following needs to be done prior to taking up for a

1. The introduction needs the latest references
2. thoroughly revision for grammatical errors is a concern to all reviewers
3. rationale behind the study and statistical significance

**Language Note:** The Academic Editor has identified that the English language must be improved. PeerJ can provide language editing services - please contact us at copyediting@peerj.com for pricing (be sure to provide your manuscript number and title). Alternatively, you should make your own arrangements to improve the language quality and provide details in your response letter. – PeerJ Staff

·

Basic reporting

The manuscript provides a comprehensive overview of research concerning alterations in protein glycosylation in meningiomas.

Experimental design

The experiments in the study were meticulously conducted; however, it appears that the research did not introduce novel findings to the existing literature.

1. The identification of highly expressed terminal galactose (Gal) and N-acetyl galactosamine (GalNAc) in primary benign meningioma cells in contrast to malignant meningioma cell lines could potentially constitute a novel observation. However, conclusive evidence necessitates further validation.

2. The mention of aberrant mucin types of O-glycosylations in meningiomas, as observed through the GEO database and mRNA experiments (COSMC, MUC1, C2GNT2, ST3GAL1, ST3GAL3, FUT1, and FUT8) in malignant meningiomas compared to benign cell lines are important.

It is essential to acknowledge the extensive literature available on these topics.

Validity of the findings

While it is evident that the authors have conducted a thorough analysis and have unveiled several significant findings, it is imperative to ascertain whether these findings represent entirely novel contributions or if they have been previously documented in the literature.

Additional comments

A few concerns require the author's attention:
Definition of Meningioma Grades: The manuscript does not clearly define meningioma cell lines into Grade I, Grade II, and Grade III.
Absence of a Grade II Cell Line: A discrepancy arises from the assertion made on lines 24 and 25 of the manuscript regarding the utilization of Grade II cell lines, as there seems to be an absence of detailed reference to this particular grade in subsequent descriptions.


Please switch the columns and rows in Figure 1 to improve the data presentation

Reviewer 2 ·

Basic reporting

Grammatical errors and factually inaccurate statements are major flaws in this manuscript.

There are a lot of grammatical errors, often resulting in scientifically inaccurate sentences. For example, while talking about the changes in protein levels, authors incorrectly talk about changes in glycosylation – protein levels and glycosylation are completely different and not interchangeable concepts.

Experimental design

The sample size is very small.

The authors do not explain the scientific rationale for any of their experimental decisions well. For example, why do authors feel Lectin cytochemistry is the best way to analyze the changes in O-glycosylation?

Validity of the findings

The study is highly biased and, therefore, doesn’t add much value to the field.


Proper statistical analyses are missing.

Additional comments

The title is misleading – the authors do not measure O-linked glycosylation. Mass spectrometry would be a better approach for such analysis. If authors wish to try to resubmit, it will require significantly more work – particularly the mass spec analysis to provide a comprehensive overview of changes in O-linked glycosylation


Please perform a literature search and make sure to include relevant papers. Elaborate how the current study is different compared to other similar papers.

Reviewer 3 ·

Basic reporting

Language used for the manuscript is clear and professional. Introduction and background are provided to describe the research topic. All figures are supplied with clear labels and descriptions. The basic reporting of the manuscript is decent.

Experimental design

First comment is that the research question is not well defined. It remains unclear whether the authors are trying to demonstrate correlation between glycosylation level and all enzymes studied in the manuscript.
Second comment is that it will be more convincing to show quantitative data to support the statement that glycosylation levels are different in benign/malignant cells.

Validity of the findings

As commented in section 'experimental design', please provide quantitative measurement of glycosylation levels in addition to the current method to support your discovery.
The causality between glycosylation and enzyme levels is weak. From this aspect, at least two more experiments need to be presented. First is the actual level of enzymes of interest by protein assay. Second is knockdown assay. Since only the overall intracellular glycosylation level has been measured, it will be very challenging to isolate related enzymes and figure out their impact. GEO data can be useful, but it is not supposed to be the most critical component of the study, you will need to add a lot more studies to support your conclusion.

---

## Round 0.2 · accepted · Accept

The paper is accepted on the reviewer recommendations

·

Basic reporting

The investigation of glycan profiles and glycosyltransferase expression in benign and malignant meningiomas contributes significantly to our understanding of the molecular changes underlying these tumors, potentially paving the way for novel diagnostic and therapeutic strategies.

Experimental design

Experiments are meticulously planned.

Validity of the findings

The study looks into the functional implications of altered glycosylation in malignant meningiomas, raising important questions about tumor behavior and immune responses. Furthermore, the clinical significance of the study, which investigates how identified glycosylation changes can be used for diagnostics and therapeutics, emphasizes its potential translational impact and aligns with current trends in personalized medicine.

Reviewer 3 ·

Basic reporting

No comment

Experimental design

No comment

Validity of the findings

No comment